# SMAD Proteins in TGF-β Signalling Pathway in Cancer: Regulatory Mechanisms and Clinical Applications

**DOI:** 10.3390/diagnostics13172769

**Published:** 2023-08-26

**Authors:** Qi Wang, Fei Xiong, Guanhua Wu, Da Wang, Wenzheng Liu, Junsheng Chen, Yongqiang Qi, Bing Wang, Yongjun Chen

**Affiliations:** 1Department of Biliary-Pancreatic Surgery, Tongji Hospital, Tongji Medical College, Huazhong University of Science and Technology, Wuhan 430000, China; eric_wang0305@outlook.com (Q.W.); sillcon2008@hotmail.com (F.X.); ghwu95@163.com (G.W.); a123131515@163.com (D.W.); liu15927464856@163.com (W.L.); cjs9800@163.com (J.C.); t0013008@aliyun.com (B.W.); 2Key Laboratory of Laparoscopic Technology of Zhejiang Province, Department of General Surgery, Sir Run-Run Shaw Hospital, Zhejiang University School of Medicine, Hangzhou 310016, China; a15972066440@163.com

**Keywords:** SMAD, transforming growth factor-β, cancer, diagnosis, treatment

## Abstract

Suppressor of mother against decapentaplegic (SMAD) family proteins are central to one of the most versatile cytokine signalling pathways in metazoan biology, the transforming growth factor-β (TGF-β) pathway. The TGF-β pathway is widely known for its dual role in cancer progression as both an inhibitor of tumour cell growth and an inducer of tumour metastasis. This is mainly mediated through SMAD proteins and their cofactors or regulators. SMAD proteins act as transcription factors, regulating the transcription of a wide range of genes, and their rich post-translational modifications are influenced by a variety of regulators and cofactors. The complex role, mechanisms, and important functions of SMAD proteins in tumours are the hot topics in current oncology research. In this paper, we summarize the recent progress on the effects and mechanisms of SMAD proteins on tumour development, diagnosis, treatment and prognosis, and provide clues for subsequent research on SMAD proteins in tumours.

## 1. Introduction

Modern medicine shows that the development of tumours is a very complex process. It is the combined result of tumour cell proliferation, metastasis, and apoptosis. A variety of growth factors and signalling proteins are involved in this process. Members of the TGF-β family control cell growth, differentiation, and apoptosis, and have important functions during embryonic development [1]. In tumours, multiple cell types produce TGF-β and respond to it, resulting in a complex network involving epithelial cells, tumour cells, immune cells, and stromal fibroblasts [2]. This complex network can cause disease and change over time, giving TGF-β both tumour suppressive and tumour promoting or enabling effects [3]. In normal non-cancerous and early cancer cells, activation of the TGF-β signalling pathway induces apoptosis and strongly inhibits the cell cycle [4,5], suggesting a major role for this signalling pathway in cancer suppression. On the other hand, the TGF-β pathway exerts tumour-promoting effects by activating multiple signalling pathways and by affecting EMT, angiogenesis and tumour invasion, and metastasis through immune evasion [6]. The key molecules that enable the TGF-β pathway to play different roles are the SMAD proteins.

The first protein of the SMAD family, an intracellular protein named Mad, was identified in the 1990s during a mutation screen for the decapentaplegic (dpp) gene responsible for the formation of Drosophila wings [7]. Subsequently, homologues of Mad proteins were identified in *Caenorhabditis elegans* and vertebrates. The term “SMAD” is derived from the combination of the gene names of two homologous proteins, Sma of *Caenorhabditis elegans* and Mad of *Drosophila melanogaster* [8]. The first SMAD to be discovered was SMAD4 or DPC4 (deleted in pancreatic cancer 4) [9].

Eight SMAD proteins are encoded by human and mouse genomes. Of these, SMAD1, SMAD2, SMAD3, SMAD5, and SMAD8 act as substrates for the TGF-β family receptor, and these proteins are commonly referred to as receptor-regulated SMADs (R-SMADs). SMAD1, 5, and 8 are mainly substrates for bone morphogenetic proteins (BMPs) and anti-Mullerian receptors; SMAD2 and 3 are principally substrates for the TGFβ, activin, and Nodal receptors. SMAD4, also known as Co-SMAD, is a cofactor for all R-SMADs. SMAD6 and 7 are inhibitory SMADs (I-SMADs) that have antagonistic effects and eliminate TGF-β signalling [10,11]. SMAD6 preferentially inhibits BMP signalling, whereas SMAD7 inhibits both the TGF-β/activin and BMP signalling pathways [12].

Structurally, the SMAD proteins consist of two spherical structural domains: the N-terminal MAD Homology domain1 (MH1), which contains a hairpin 4 structure with DNA binding capacity, and the C-terminal MAD Homology domain2 (MH2), which contains a hydrophobic element that binds to the transmembrane receptors TGF-βR and BMPR. The intermediate linker region is a flexible fragment with post-translational modification sites such as the binding site for SMURF (SMAD ubiquitination-related factor) ubiquitin ligase and the phosphorylation site for protein kinase such as TGF-β type I receptor [10].

SMAD family proteins function primarily in the TGF-β pathway. TGF-β family ligands initiate downstream signalling events by activating transmembrane serine-threonine kinase receptors, namely, the TGF-β type I receptor (TbRI), which is also named as activin receptor-like kinase 5 (ALK5), and the TGF-β type II receptor (TbRII). On ligand binding, TbRII forms a heterotetramer with TbRI/ALK5, and the glycine/serine-rich (GS) structural domain of TbRI/ALK5 is phosphorylated by TbRII, which is a 30 amino acid region rich in glycine and serine that precedes the kinase structural domain. Anchoring proteins capture R-SMADs for presentation to activated type I receptors [13,14]. Inactive SMADs are in an inhibitory state through intermolecular interactions between the MH1 and MH2 structural domains [15]. The MH2 structural domain of R-SMADs is phosphorylated in the presence of activated type I receptors, thereby relieving their inhibitory state. Activated R-SMADs bind to Co-SMAD and SMAD4 to form a polymer that completes the SMAD4-mediated plasmid-nucleus shuttle. Because of the low binding affinity of the SMAD complex for DNA, other transcription factors are required for high-affinity interactions and chromatin specificity. After entering the nucleus, the SMAD complex recruits co-activators such as P300/CBP [16,17] and GCN5 [18] to exert transcriptional activation. At the same time, it can also recruit co-repressors such as TGIF [19], Ski [20,21], and SnoN [22] to exert transcriptional repression. Meanwhile, the I-SMADs, especially SMAD7, inhibit SMAD-mediated signalling by competing with activated R-SMADs to bind SMAD4 or interacting with upstream receptors (Figure 1) [23].

The dual role of TGF-β in cancer has long been identified, but its mechanistic basis, operational logic, and clinical relevance are still unclear. In this paper, we summarize the recent progress in the research and application of SMAD proteins in tumours, in order to find the contribution of SMAD proteins to the dual role of TGF-β and provide clues for the subsequent research on SMAD proteins in tumours.

### 1.1. Oesophageal Cancer

Barrett’s oesophagus (BE) is formed when squamous epithelium is replaced by precancerous tissue (dysplasia) and is associated with an increased risk of developing oesophageal adenocarcinoma (EAC) [24]. Studies have shown that SMAD family proteins and the TGF-β signalling pathway play an important role in the progression of Barrett’s oesophagus to oesophageal cancer.

The percentage of linker threonine-phosphorylated SMAD2/3 (pSMAD2/3L-Thr) positive cells increases significantly with the progression of oesophageal tumours. In low-grade intraepithelial tumours, pSMAD2/3L-Thr-positive cells were distributed in the lower segment of low-grade intraepithelial neoplasia, and were observed up to the upper section in carcinoma in situ. In invasive squamous cell carcinoma, they were scattered throughout the tumour with a loss of polarity and were found in primary intraepithelial lesions and at sites of submucosal and vascular invasion. pSMAD2/3L-Thr was significantly expressed in human oesophageal non-tumour and tumour epithelium, suggesting that they are epithelial and cancer stem cells associated with oesophageal tumourigenesis, respectively [25]. Furthermore, SMAD4 expression was significantly reduced in all stages of Barrett’s oesophagus from chemotaxis to hypo- and highly heterogeneous hyperplasia [26], and knockdown of SMAD4 was sufficient to promote tumourigenesis in dysplastic Barrett’s oesophagus cells in vivo [27]. Heterozygous deletions of the region on chromosome 18q containing SMAD2 and SMAD4 were present in approximately 30–70% of patients with precancerous BE [28] and in approximately 70% of oesophageal adenocarcinomas associated with BE [28,29], suggesting that such heterozygous deletions are early events in tumour transformation.

In oesophageal cancer, ATP-binding cassette, sub-family B (MDR/TAP), member 7 (ABCB7) knockdown inhibits TGF-β signalling pathway transduction by promoting SMAD7 expression and repressing SMAD3 expression, inducing apoptosis and suppressing EMT in oesophageal cancer cells [30]. The transcription factor forkhead box D3 (FOXD3) can bind directly to the promoter region of the SMAD7 gene, leading to the transcriptional promotion of SMAD7 in human oesophageal cancer cells, inhibition of the TGF-β pathway, and suppressive effects in oesophageal squamous cell carcinoma [31]. It was shown that SMAD7, as a repressor molecule of the TGF-β pathway, is underexpressed in oesophageal carcinoma through a variety of genes, allowing for abnormal activation of the TGF-β pathway and promoting the progression of oesophageal carcinoma.

### 1.2. Gastric Cancer

Gastric cancer is one of the most common malignant tumours in the world and has a high morbidity and mortality rate. Although significant progress has been made in the treatment of gastric cancer, the 5-year survival rate is only 20–30% due to factors such as the lack of specific clinical manifestations and targeted drugs in the early stages [32].

In gastric cancer, ubiquitin-related enzymes such as TRIM22 [33] and USP32 [34] promote the proliferation and invasion of gastric cancer cells by affecting the stability of the SMAD2 protein, which is highly expressed in gastric cancer. SMAD2 expression levels are low in EBV-associated gastric cancer, particularly in the presence of EBV-encoded latent membrane protein 2A (LMP2A). It has been shown that LMP2A promotes miR-155-5p expression through the activation of nuclear factor-κB (NFκB) signalling, and the overexpression of miR-155-5p inhibits SMAD2. Immunofluorescence analysis further showed that LMP2A blocks p-SMAD2 translocation to the nucleus. Thus, the role of LMP2A in EBV-positive GC may lead to a favourable prognosis by promoting apoptosis and cell cycle arrest as well as inhibiting tumour proliferation [35]. The above studies suggest that SMAD2, an important mediator of the TGF-β pathway, plays an important role as an oncogene in the development of gastric cancer. In addition, a variety of microRNAs influence the development of gastric carcinogenesis by affecting the levels of the SMAD family of proteins, as detailed in Table 1.

In terms of tumour treatment, different treatment strategies are associated with SMAD expression. Surgery alone increased the expression levels of SMAD1, SMAD2, and SMAD4. Conversely, treatment with 5-FU-based adjuvants decreased the expression levels of SMAD3 and SMAD6, but increased the expression of SMAD5. In addition, high levels of SMAD9 expression were associated with adverse effects in patients treated with other adjuvants [36]. The expression levels of SMAD proteins have an important impact on the tumour microenvironment and therapeutic efficacy. Studies have shown that SMAD1 is highly expressed in cisplatin-resistant gastric cancer cells and that SMAD1 interacts with YAP1, leading to the increased resistance of gastric cancer cells to cisplatin [37].

The introduction of immunotherapy, represented by immune checkpoint blockade, has brought about a turnaround in the treatment of gastric cancer. However, clinical studies have found that the efficacy of this treatment varies greatly between individual patients. Recently, the impact of SMAD family protein expression levels on immunotherapy has been increasingly recognised. It was shown that SMAD4 deletion allowed gastric cancer cells to evade tumour immunity. SMAD4-deficient GC cells exhibited the expansion of CD133+ cancer stem cells, along with the inhibition of dendritic cell (DC) differentiation and the aggregation of cytotoxic T cells with granulocyte myeloid-derived suppressor cells (G-MDSC) via CXCL1-containing secretomes. In addition, SMAD4 deletion increased programmed cell death ligand-1 (PD-L1) and decreased 4-1BBL expression, indicating altered immunogenicity. Combined immune checkpoint blockade (ICB) with anti-PD-L1 and anti-CTLA-4 or agonistic anti-4-1BB antibodies effectively treats ICB monotherapy-resistant SMAD4-deficient allografts, providing a rational basis for an ICB strategy to treat advanced SMAD4-deficient GC [38].

Due to the important role of SMAD family proteins in the development of gastric cancer and response to treatment, SAMD protein levels are closely related to the prognosis of gastric cancer patients. Studies have shown that higher levels of SMAD1, SMAD2, and SMAD4 expression are associated with good overall survival (OS) in stage I and II cancer. On the other hand, increased expression of SMAD3, SMAD5, SMAD6, and SMAD7 was associated with low OS in stage I and II of the cancer. In all gastric cancers, increased SMAD9 expression was associated with poor OS [36]. In a subgroup analysis based on tumour node metastasis (TNM) stage, SMAD4 and SMAD7 showed the most significant prognostic differences in patients with stage I gastric cancer [39]. Based on previous studies, it is known that the TGF-β pathway promotes the EMT process in tumours and is often associated with a poorer prognosis. However, the above data suggest that activation of the TGF-β pathway improves the prognosis of some patients. Therefore, the complexity of whether TGF-β signalling is tumour-suppressive or tumour-promoting remains to be further investigated.

### 1.3. Colorectal Cancer

The role of SMAD proteins in colorectal cancer is more complex and well-studied. In CRC development, as early as 1998, Zhu et al. found that mice with a full knockout of SMAD3 could spontaneously develop CRC [40]. More recently, Gu et al. showed that mice heterozygous for SMAD4 and the gene encoding the SMAD3 adaptor protein SPTBN1 also developed CRC spontaneously and that these mice exhibited altered gut microbiota that resembled that associated with human CRC [41]. Furthermore, BRAF-driven colorectal cancer is one of the most poorly prognosed subtypes of colorectal cancer. In the oncogenic BRAF-V600E mouse model, deletion of the tumour suppressor SMAD4 promoted rapid development and the progression of serrated tumours, and SMAD4 mutations co-occurred with BRAF-V600E mutations in human patient tumours [42]. These findings identify SMAD4 as a key factor in early-stage serrated cancers and help to expand the understanding of this rare but aggressive subgroup of colorectal cancers.

For the progression of CRC, SMAD4 plays an important role. A growing body of evidence confirms that SMAD4 is lost in colorectal cancer at a frequency of approximately 30% [43]. Inactivation of SMAD4 leads to increased secretion of a variety of proteins known to be involved in pro-metastatic processes. For example, it has been shown that DKK3, one of the factors secreted organ-specific by SMAD4 mutants, reduces the antitumour effects of natural killer cells (NK cells) [44]. TRIM47 promotes the expression of CCL15 by promoting SMAD4 ubiquitination and degradation, and promotes the growth and invasion of human CRC cells through CCL15-CCR1 signalling [45]. Furthermore, in colorectal cancer, the ribosomal biogenesis factor NLE1 plays a key role in tumour growth and progression. In the absence of SMAD4, the TGF-β signalling pathway-mediated downregulation of NLE1 is prevented by the ectopic expression of c-MYC, which occupies the E-box-containing region within the NLE1 promoter and upregulates NLE1, thereby promoting tumour progression [46]. The above studies suggest that decreased expression of SMAD4 has a facilitative effect on CRC progression.

However, SMAD4 does not completely inhibit CRC progression. Li et al. found that SMAD4 in combination with SMAD3 could positively regulate the vascular endothelial growth factor C (VEGF-C) during colon cancer metastasis by binding to the promoter of the VEGF-C gene, which is essential for invasive metastasis in CRC. However, at the same time, SMAD4 increased the transcription of miR-128-3p, a microRNA targeting VEGF-C mRNA, leading to the downregulation of VEGF-C expression. Ultimately, Li et al. found that the long non-coding RNA (lncRNA) ASLNC07322, which is specifically increased in metastatic colon cancer, acts as a sponge for miR-128-3p to reduce it, leading to a subsequent increase in VEGF-C. ASLNC07322 critically controls this negative and positive regulatory transition between them, which in turn balances the SMAD4 effects on CRC invasive metastasis [47].

The role of the SMAD family of proteins on the immune microenvironment of tumours and the tumour immune response has also received considerable attention in recent years. Hanna et al. found that the regulation of mucosal inflammation in ulcerative colitis-associated tumours was central to the tumour suppressive function of SMAD4 in the colon. SMAD4 deficiency in the mouse colonic epithelium leads to an expansion of gut-associated lymphoid tissue and the recruitment of immune cells to the mouse colonic epithelium and stroma, particularly T regulatory cells, Th17, and dendritic cells. A key downstream node of this regulation is the inhibition of epithelial chemokine c-c motif chemokine ligand 20 (CCL20) signalling to CCL20/c-c motif chemokine receptor 6 (CCR6) in immune cells. Deletion of SMAD4 in colonic epithelial cells increases CCL20 expression and chemotaxis of CCR6+ immune cells, contributing to increased susceptibility to colon cancer [48]. In addition, interferon-γ cell expression was significantly increased in T cells and colonic mucosal epithelium of mice deficient in SMAD4, and increased IFN-γ expression promoted colorectal carcinogenesis through immunomodulatory mechanisms and directly on endothelial and epithelial homeostasis. SMAD4 knockdown also upregulated CXCL1 and CXCL8 expression, recruiting neutrophils into colorectal tumours. Both CXCL1 and CXCL8 were abundantly expressed in tumour-infiltrating neutrophils. Statistical analysis showed that CRC patients with high levels of CXCL8 exhibited shorter OS and recurrence-free survival (RFS) [49]. The above studies suggest that SMAD4 deficiency contributes to colorectal cancer development by affecting the response of immune cells.

Given the important role of SMAD4 in the development of CRC, SMAD4 has a significant indicative role in the prognosis of CRC patients. Analysis of a cohort of 364 patients with stage I–IV CRC showed that SMAD4 deficiency was associated with higher tumour and nodal staging, the use of adjuvant therapy, fewer tumour infiltrating lymphocytes (TIL), lower peritumour lymphocyte aggregation (PLA) scores, and poorer RFS. Among patients receiving 5-fluorouracil (5-FU)-based systemic chemotherapy, the median RFS was 3.8 years in SMAD4-deficient patients compared to 13 years in SMAD4-preserved patients [50]. In conclusion, SMAD4 deficiency was associated with poorer clinical prognosis, chemoresistance, and reduced immune infiltration, supporting its use as a prognostic marker in cancer patients. An analysis of the prognostic and predictive value of actionable mutated genes in metastatic colorectal cancer (mCRC) revealed that concurrent mutations in TP53 and SMAD4 were associated with an increased risk of death (*p* = 0.03; HR:2.91). In mCRC patients treated with first-line regimens, SMAD4 mutations in TP53-altered tumours predicted negative prognostic outcomes [51]. In addition, mutational status of SMAD4 had a role in predicting prognosis after resection of liver metastases from colorectal cancer [52].

For the diagnosis of CRC, SMAD3 has a more prominent role. In a Taiwanese cohort study, investigators found hypomethylated SMAD3 in 91.4% (501/548) of Taiwanese colorectal cancer tissues and 66.6% of benign tubular adenoma-polyp tissues. In addition, SMAD3 hypomethylation was observed in 94.7% of CRC patients in the Cancer Genome Atlas dataset. A reduction in the methylation of SMAD3 in circulating cell-free was detected in 70% of CRC patients, but in only 20% of healthy individuals. SMAD3 mRNA expression was low in 42.9% of Taiwanese CRC tumour tissues but high in 29.4% of tumours compared with paired adjacent normal tissues. These results suggest that SMAD3 hypermethylation is a potential diagnostic marker for CRC in Western and Asian populations [53]. Another study found that the microRNA-375 and rs4939827 SNPs in SMAD7 could be considered as potential markers for the detection and early diagnosis of CRC patients [54].

SMAD3 is also a key molecule in the chemoresistant phenotype of CRC. In a group of 76 patients with locally advanced rectal cancer (LARC), the expression of SMAD3 and phosphorylated SMAD3 in preoperative tumour tissues was assessed by immunohistochemistry, and SMAD3 polymorphisms (rs35874463, rs1065080, rs1061427, rs17228212, rs744910, and rs745103) in relation to tumour regression grade and patient prognosis. The results showed that patients with high tumour expression of SMAD3 were at significantly increased risk of poor response to neoadjuvant chemotherapy. Patients carrying the variant SMAD3 rs745103-G allele had a slightly poorer response (OR:0.48, *p* = 0.0093), longer OS (HR:0.65, *p* = 0.0307), and a trend towards prolonged progression-free survival (HR:0.75, *p* = 0.0944). Patients carrying both high SMAD3 tumour expression and the wild-type rs745103-A allele had an extremely high risk of failing to achieve a complete response (OR:13.45, *p* = 0.0005). Host and tumour SMAD3 status may be considered to improve the risk stratification of LARC patients to facilitate the selection of other personalised neoadjuvant treatment strategies including intensive treatment regimens [55]. In addition, Mattia et al. found that patients with the SMAD3rs7179840-C allele present in CRC tumours had a higher OS rate after treatment with FOLFIRI (irinotecan, 5-FU, folinic acid). This finding may provide a new decision-making tool for improving the clinical management of CRC patients receiving FOLFIRI [56].

For SMAD7, which has a dual role in CRC progression, it promotes tumourigenicity in non-tumourigenic CRC cell lines by inhibiting the TGF-β signalling pathway [57]. In addition, CRC cells contain high levels of active signal transducers and transcriptional activators (STAT)-3 that exert proliferative and anti-apoptotic effects. There is a positive correlation between SMAD7 and STAT3, promoting STAT3 expression and acting as a promoter of CPC progression [58]. In contrast, the m1 A demethylase alkB homolog 1 (ALKBH1) promoted CRC metastasis by promoting methyltransferase 3, N6-adenosine-methyltransferase (METTL3), which destabilised SMAD7 [59]. This suggests that SMAD7 plays an inhibitory role in CRC progression. In conclusion, the SMAD protein family, which provides important indications for the early diagnosis of CRC and the choice of treatment options, offers great clinical value.

### 1.4. Hepatocellular Carcinoma

The role of the TGF-β pathway in hepatocarcinogenesis and development is paradoxical. Previous studies have shown that inhibition of the TGF-β pathway appears to have a promotional effect on hepatocarcinogenesis. Baek et al. found that altered SMAD3 function occurring in mice heterozygous for the SMAD3 bridging protein SPTBN1 led to fibrosis and spontaneous development of hepatocellular carcinoma (HCC), which was associated with overproliferating endothelial cells, leading to abnormal angiogenesis [60]. Another study in cultured hepatocyte progenitor cells found that inhibition of SMAD3-mediated gene expression downstream of TLR4 contributed to the induction of cancer stem cell properties [61]. The above experiments suggest that alterations in SMAD3, a key protein of the TGF-β pathway, have an important role in the natural occurrence of HCC.

However, more experiments have found that activation of the TGF-β pathway has a stronger role in promoting hepatocarcinogenesis. Wu et al. identified an oncogenic lncRNA that is upregulated in HCC and is transcriptionally induced by TGF-β (named lnc-UTGF). Lnc-UTGF interacts with SMAD2 and SMAD4 mRNAs through complementary base pairing, enhancing the stability of SMAD2/4 mRNA and thus promoting the TGF-β signalling pathway, while SMAD2 /3 binds to the lnc-UTGF promoter and activates lnc-UTGF-expression, thus forming a TGF-β/SMAD/lnc-UTGF positive feedback loop and promoting the invasive migration of HCC [62]. In addition, SEPHS1 promotes HCC migration and invasion by promoting the expression of SMAD2/3/4 [63]. STMN2 disrupts the microtubule–SMAD complex, promotes the release of SMAD2/3 from the microtubule network and phosphorylates it, promoting epithelial mesenchymal transformation in HCC [64]. BUB1 promotes HCC proliferation by inducing phosphorylation of SMAD2 [65]. SMAD3 binds to the protein tyrosine phosphatase receptor ε (PTPRε) promoter to activate its expression and, in turn, PTPRε interacts with TGF-βR1/SMAD3 to promote the recruitment of SMAD3 to TGF-βR1, resulting in a sustained state of SMAD3 activation that promotes HCC cell migration, invasion, and metastasis in vitro and in vivo [66]. SMAD4 is highly expressed in HBV-positive hepatocellular carcinoma patient samples, promotes HCC cell proliferation, and is associated with poor prognosis. Mechanistically, HBx regulates SMAD4 expression at the transcriptional level via TFII-I and can bind to SMAD4 to inhibit its ubiquitination. SMAD4 can also promote the expression of HBx through a positive feedback mechanism [67]. All of the above studies suggest that various oncogenes can promote HCC progression by promoting the activation or expression of SMAD2/3 and activating the TGF-β pathway.

It was also found that mice with SMAD7 hepatocyte-specific overexpression showed protection against chemically induced fibrosis and liver injury [68], while mice with SMAD7 liver-specific deletion showed increased susceptibility to chemically induced HCC [69]. ATXN7L3 also promotes SMAD7 transcription through the regulation of histone H2B ubiquitination levels and is subsequently involved in the inhibition of tumour growth in HCC [70]. The above experiments also suggest that SMAD7 can exert an inhibitory effect on HCC development by inhibiting the TGF-β pathway. In summary, most experiments demonstrated the promoting effect of the TGF-β pathway on hepatocellular carcinoma, however, the inhibitory effect exhibited by the TGF-β pathway in a few experiments cannot be ignored, showing the complex and critical role of the TGF-β pathway in tumours, and SMADs, as key mediators of the TGF-β pathway, are an important entry point for future studies of the TGF-β pathway.

Chinese medicine has always played an important supporting role in the treatment of HCC, and has been effective in controlling tumour progression, improving clinical efficacy, reducing adverse effects, and preventing recurrence and metastasis at different stages of HCC when combined with Western medicine. SMAD3 phosphorylation is associated with hepatic fibrous carcinogenesis. pSMAD3L and pSMAD3C, the isomers of SMAD3 phosphorylation, are reversible and antagonistic, and the balance may shift from carcinogenesis to tumour suppression. pSMAD3C has recently been assigned a preventive role against primary liver injury. Salvianolic acid B can delay the progression of hepatic fibrosis-associated carcinoma by converting pSMAD3L/3C in mice and is a key target for the prevention of hepatocarcinogenesis [71]. In addition, SMAD7, a key molecule in anti-liver fibrosis, is the target of several herbal medicines. Citicoline has a protective activity against liver by upregulating SMAD7, inhibiting the TGF-β pathway and suppressing the activation of hepatocyte EMT and haematopoietic stem cells [72]. Jianpi Huayu Decoction (JPHYD) is an ancient traditional Chinese medicine formula. Liu et al. found that JPHYD promotes the expression of SMAD7 by inhibiting miR-21-5p, which in turn exerts an inhibitory effect on the proliferation, invasion, and migration of HCC cells [73]. Compound kushen injection (CKI), a typical traditional Chinese medicine (TCM), has been approved to clinically treat cancer-induced pain in China for over 20 years by the Chinese National Medical Products Administration (NMPA). CKI was found to inhibit hepatic stellate cell (HSC) activation by stabilising SMAD7/TGF-βR1 interactions to rebalance SMAD2/SMAD3 signalling and reduce extracellular matrix formation. SMAD7 deletion abrogated the antifibrotic effects of CKI in vivo and in vitro, suggesting that SMAD7 is an important target for CKI antifibrosis [74].

In cholangiocarcinoma, next-generation sequencing of 31 cholangiocarcinoma cases identified SMAD4 as a favourable prognostic biomarker in intrahepatic CCA and perihilar CCA by univariate and multivariate analysis. SMAD4 inhibited the proliferation, migration, and invasion of CCA, and enhanced the sensitivity of cholangiocarcinoma to pemotinib, the only FDA-approved targeted drug for CCA by inhibiting β-catenin-S675 phosphorylation and intranuclear translocation [75].

### 1.5. Pancreatic Cancer

Pancreatic cancer (PC) is one of the most difficult human cancers to treat, with ductal adenocarcinoma of the pancreas (PDAC) accounting for approximately 85% of diagnosed cases. The OS of pancreatic cancer is reported to be very poor, with a median survival of only 7–8 months and 1-year and 5-year survival rates of 35.0% and 4.4%, respectively [76]. PDAC usually follows a predictable morphological and genetic process that transforms normal epithelial cells into non-invasive pancreatic intraepithelial neoplasia (PanIN), which in turn develops into PDAC. In this genetic process, the presence or absence of SMAD4/DPC4 plays a crucial role in the development, treatment, and prognosis of PDAC.

Since the discovery of SMAD4/DPC4 in 1996, many studies have shown that alterations in SMAD4/DPC4 are closely associated with pancreatic cancer. SMAD4/DPC4 deletions occur late in the transformation of pancreatic intraepithelial neoplasia to pancreatic cancer [77], with heterozygous deletions occurring in nearly 60% of human pancreatic cancers and approximately 50% exhibiting homozygous deletions or intragenic inactivating mutations [9]. Further studies have shown that SMAD4/DPC4 mutations were associated with pancreatic pathological staging. A total of 31% (9/29) of cases had SMAD4/DPC4 inactivated in high-grade tumours (Pan in-3), while no inactivation was found in the remaining 159 low-grade lesions (Pan in-1 and 2) [77]. Pancreas-specific PDX1-Cre or P48-Cre knockdown of SMAD4/DPC4 significantly promoted KRASG12D [78] activation or PTEN [79] inactivation, triggering tumour progression. In addition, SMAD4/DPC4 deletion induced the upregulation of the glycolytic enzyme PKG1 [80] and the leucine zipper protein FOSL1 [81], enhancing glycolysis and aggressive tumour behaviour and promoting lung metastasis. In a cohort of PDAC patients, SMAD4/DPC4-negative tumours with high levels of phospho-SMAD2 were more aggressive and had a poorer prognosis. Loss of SMAD4/DPC4 tumour suppressive activity in PDAC leads to the acquisition of oncogenic function of SMAD2/3 and the associated deleterious effects [82]. These previous studies suggest that SMAD4/DPC4 has a significant tumour suppressive function in the progressive phase of PC.

Although SMAD4/DPC4 deficiency in tumour cells enhanced proliferation in vitro, in vivo growth of SMAD4/DPC4-deficient PDAC tumours was significantly inhibited in immunoreactive C57BL/6 (B6) mice. SMAD4/DPC4 deficiency significantly increased tumour cell immunogenicity by promoting spontaneous DNA damage and stimulating STING-mediated type I interferon signalling, which contributed to the activation of type 1 conventional dendritic cells (cDC1) and subsequently CD8+ T cells for tumour control. Thus, SMAD4/DPC4 deficiency promotes PDAC immunogenicity through the induction of type I interferon signalling triggered by intrinsic tumour DNA damage [82]. This study suggests that SMAD4/DPC4 is not an absolute tumour suppressor in PDAC but rather has a positive effect on the immune microenvironment of tumours. Wang et al. established a SMAD4/DPC4-driven immune signature (SDIS) in ICGC-AU2 (microarray data) through a machine learning algorithm, which was validated by database analysis, and multivariate Cox regression showed that SDIS can be a powerful predictor of prognosis in PC and can be used to further adjust chemotherapy and immunotherapy decisions in the clinical setting [83].

It is known from previous studies that not all PDACs exhibit a SMAD4/DPC4 deficiency. A comparative study of SMAD4/DPC4-negative and SMAD4/DPC4-positive tumours revealed that the presence or absence of SMAD4/DPC4 had a differential impact on the biological behaviour of pancreatic cancer. In PDAC, the effect of TGF-β1-induced autophagy on tumours was dependent on the alteration of SMAD4/DPC4. In SMAD4/DPC4-positive PDAC cells, TGF-β1-induced autophagy promoted proliferation and inhibited migration by reducing the nuclear translocation of SMAD4/DPC4. In contrast, TGF-β1-induced autophagy inhibited proliferation and promoted migration in SMAD4/DPC4-negative cells by regulating MAPK/ERK activation [84]. Using time-lapse microscopy to observe the invasion of 25 surgically excised human PDAC samples in collagen I, PDAC organisms exhibited two distinct, morphologically defined invasive phenotypes when cultured in collagen I, namely, the mesenchymal and collective. It was shown that the differences in PDAC invasion phenotypes were dependent on SMAD4/DPC4 status, with the collective invasion only present in SMAD4/DPC4-deficient PDAC. This class of organ models highlights the importance of SMAD4/DPC4 deletion in PDAC invasion, suggesting that the invasive processes of the SMAD4/DPC4 mutant and SMAD4/DPC4 wild-type tumours differ in both morphological and molecular mechanisms [85].

In light of the important role of SMAD4/DPC4 in PDAC, a large number of drugs targeting SMAD4/DPC4 as a therapeutic target are currently being explored. Yao et al. developed an EGFR/HER2-targeting conjugate, the dual-target ligand-based lidamycin (DTLL), which prevents tumour proliferation in SMAD4/DPC4-deficient PDAC through ATK/mTOR blockade and impaired NF-κB function, and also restores SMAD4/DPC4 bioactivity, triggering downstream NF-κB regulation in SMAD4/DPC4-deficient tumours of signalling to overcome chemoresistance. In combination with gemcitabine, DTLL exhibits a unique mechanism of action similar to that of SMAD4/DPC4, showing superior inhibition of the growth of gemcitabine-resistant/sensitive tumours [86]. Fei et al. found that the addition of hydroxychloroquine to neoadjuvant chemotherapy in patients with PDA may improve the therapeutic response in SMAD4/DPC4-deficient patients. Furthermore, in PDAC, SMAD4/DPC4 deficiency induced the overexpression of hepatic nuclear factor 4γ (HNF4G), which correlated with poor metastasis and survival time in xenograft animal models and PDAC patients (log rank *p* = 0.036; HR = 1.60, 95% CI = 1.03–2.47). Metformin inhibited HNF4G activity and improved the clinical outcomes and survival in patients with SMAD4/DPC4-deficient PDAC (log rank *p* = 0.022; HR = 0.31, 95% CI = 0.14–0.68), but not in patients with SMAD4/DPC4-normal PDAC [87]. However, Ezrova et al. found that SMAD4/DPC4 deficiency promoted increased MAPK/ERK signalling-driven mitotic flux and thus induced resistance to biguanides. In contrast, mitochondrial targeting of tamoxifen, a complex I inhibitor in clinical trials, overcame resistance mediated by SMAD4/DPC4 deficiency or TGF-β signalling. These findings could help in the development of mitochondria-targeted therapies for pancreatic cancer patients with SMAD4/DPC4 as a reasonable predictive marker [88]. In summary, due to the lack of SMAD4/DPC4 in most PDACs, the development of drugs that can function as an alternative to SAMD4 has become a popular target for pancreatic cancer therapy. However, further breakthrough studies are still needed for the therapeutic target of SMAD4/DPC4-positive pancreatic cancer. In addition, more clinical studies are still needed to explore whether biguanide drugs are effective in the treatment of PDAC patients with SMAD4/DPC4 deficiency.

In addition to SMAD4/DPC4, other SMAD family proteins also have an effect on pancreatic cancer. For example, ITGA2 inhibits the activation of the TGF-β pathway through transcriptional repression of SMAD2 expression, which in turn promotes the growth of pancreatic cancer cells [89]. KLF16 promotes tumour proliferation and invasion by promoting the expression of SMAD6 [90]. Nuclear IL-33/SMAD signalling is a cell-autonomous tumour-promoting axis in chronic inflammation. IL-33 suppresses SMAD6 expression and promotes phosphorylated SMAD1/2/3/5 expression, which is upregulated in skin cancer and pancreatitis-associated pancreatic cancer [91]. Honokiol inhibits the invasive migration and perineural invasion of pancreatic cancer by inhibiting the phosphorylation of SMAD2/3, reducing the tumour cell damage to the sciatic nerve and protecting sciatic nerve function [92]. In addition, high SMAD2 expression [93] and low SMAD7 expression [94] were negatively associated with OS in PC patients. Based on the above studies, it is evident that the TGF-β pathway has equally contradictory and complex effects on pancreatic cancer. SMAD family proteins, as key molecules of the TGF-β pathway, need to be studied more in order to explore the regulatory points of the TGF-β pathway’s action on tumours.

### 1.6. Lung Cancer

Lung cancer is the leading cause of cancer-related deaths worldwide. Non-small cell lung cancer (NSCLC) comprises most histological subtypes of lung cancer including adenocarcinoma and squamous cell carcinoma. The TGF-β pathway is significantly activated in NSCLC and plays an important role in the development of NSCLC.

Unlike CRC and PC, SMAD3 has been studied much more than SMAD4 in NSCLC. USP7 promotes positive self-regulation of SMAD3 by catalysing SMAD3 deubiquitination and inhibits the progression of p53-deficient lung cancer, suggesting that SMAD3 can play an anti-cancer role in lung cancer [95]. In lung epithelial cells, SMAD3 binds to ATOH8 to form a transcriptional complex that directly represses a series of cell cycle-promoting genes, leading to senescence and tumour suppression in lung epithelial cells. The deletion of ATOH8 is one of the key mechanisms underlying the transition of SMAD3 from cancer suppression to cancer promotion [96]. This finding provides clues to the mechanisms underlying the dual role of the TGF-β pathway in tumours. More studies have demonstrated the pro-cancer role of SMAD3. In lung cancer cell lines, histone methyltransferase SMYD2 promoted invasive metastasis by reducing the distribution of H3K4me1 in the SMAD3 promoter region and promoting SMAD3 expression [97]. The stent protein PDLIM5 (PDZ and LIM structural domain protein 5, ENH) is a novel regulator of SMAD3 stability and critically promotes TGF-β signalling and tumour progression by maintaining SMAD3 stability in NSCLC [98]. By inhibiting SMAD3 expression, ALKBH5 promotes SMAD6 expression and suppresses TGF-β pathway signalling, which in turn inhibits NSCLC cell invasion [99]. Myocardin and SMAD3/SMAD4 form a positive feedback loop that drives TGF-β-induced EMT in NSCLC cancers [100]. The above studies suggest that multiple oncogenic factors exert pro-cancer effects through SMAD3. SMAD3 also promotes lung cancer progression by influencing downstream factors. SMAD3 binds to the RAB26 promoter, promoting its expression and promoting NSCLC progression [101]. SMAD3 knockdown significantly inhibits lung adenocarcinoma cell growth and increases radiosensitivity via P21. This suggests that SMAD3 is a potential prognostic and radiosensitivity indicator and a target for radiotherapy and other treatments for lung adenocarcinoma patients [102].

In addition, SMAD3 drives lung carcinogenesis by affecting tumour-associated fibroblasts (TAFs). TAFs play an important role in tumour microenvironment (TME)-driven cancer progression, and in lung adenocarcinoma-associated fibroblasts, SMAD3 expression is elevated, promoting fibroblast migration. In addition, SMAD3 drives lung carcinogenesis by affecting tumour-associated fibroblasts. It was found that macrophage-myofibroblast transformation (MMT) has a novel role in the de novo generation of native TAFs in cancer. Mechanistically, a regulatory network centred on SMAD3 is upregulated in MMTs in NSCLC, and chromatin immunoprecipitation sequencing (ChIP-seq) detected that SMAD3 is predominantly bound in fibroblast differentiation-related genes in macrophage lineage cells in Lewis lung carcinoma, and that macrophage-specific deletion and pharmacological inhibition of SMAD3 can effectively block MMT, thereby inhibiting TAF formation and cancer progression in vivo. Therefore, SMAD3 and MMT may be a new therapeutic target for cancer-associated fibroblasts for cancer immunotherapy [103]. In addition, Ikemori et al. reported a clinical benefit of the antifibrotic drug nintedanib in adenocarcinoma (ADC) but no significant clinical benefit in squamous cell carcinoma (SCC) through a trial of nintedanib in NSCLC. However, in practice, tumour fibrosis was actually higher in ADC-TAFs than in SCC-TAFs in both in vitro and patient samples. It was shown that the reduced fibrosis and nintedanib response in SCC-TAFs compared to ADC-TAFs was associated with increased promoter methylation of SMAD3. Due to smoking and the anatomical location of the SCC in the proximal airways, which allows for greater exposure to cigarette smoke particles, cigarette smoke condensates selectively increase SMAD3 promoter methylation, exposing SCC-TAFs to stronger epigenetic suppression of SMAD3. This caused a compensatory increase in TGF-β1/SMAD2 activation. By knocking down SMAD3 in ADC-TAFs similarly increased TGF-β1/SMAD2 activation and reduced their fibrotic phenotype and anti-tumour response to nintedanib. This result suggests that the histotype-specific regulation of lung cancer tumour fibrosis is mediated by differential levels of SMAD3 promoter methylation in TAFs. It also suggests that the target of action for the antifibrotic effect of nintedanib is SMAD3, and that ADC patients may benefit from antifibrotic agents targeting stromal TGF-β1/SMAD3 [104].

Meanwhile, SMAD3 can also influence lung cancer progression by affecting the tumour immune microenvironment. In tumours, neutrophils are dynamic, and their phenotype and function are determined by the microenvironment such as the N1 anti-tumour and N2 pro-tumour states in the tumour microenvironment (TME). It was found that among the tumour-associated neutrophils (TANs) from NSCLC patients, activation of SMAD3 in N2 TANs was negatively correlated with N1 numbers and patient survival. In an experimental lung cancer mouse model, the TAN in wild-type mice is predominantly in the N2 state, while in SMAD3-KO mice, it shifts to the N1 state, which is associated with enhanced neutrophilic infiltration and tumour regression. Mechanistically, the tumour microenvironment induced SMAD3 expression to maintain the TAN in the N2 state. Gene deletion and pharmacological inhibition of SMAD3 enhances the anticancer capacity of neutrophils against NSCLC by promoting their N1 maturation [105]. Thus, the study suggests that SMAD3 signalling in neutrophils may be a therapeutic target for cancer immunotherapy. Furthermore, phosphorylation levels of SMAD3 in CD3, CD8, Foxp3, and CD68 cells in non-small cell lung cancer negatively impact the overall and partial disease-free survival in lung cancer patients, independent of histological subtype. The high frequency of positive SMAD3 phosphorylation in Foxp3-regulatory T cells near CD8+ T cells within the tumour identifies a rapidly progressive lung cancer patient population [106]. In summary, SMAD3, one of the key mediators of the TGF-β pathway, plays a crucial role in lung cancer progression by influencing the tumour microenvironment or assisting other oncogenes. It also shows that SMAD3 has a promising future as a therapeutic target for lung cancer.

In addition to SMAD3, SMAD4 and SMAD7 are also involved in a range of biological behaviours in lung cancer. In lung cancer, loss of SMAD4 function accelerates metastasis, and it was found that SMAD4 mediates metastatic signalling through the PAK3-JNK-Jun pathway. miR-495 and miR-543, which are dependent on SMAD4, decrease in expression after loss of SMAD4 function. These miRNAs bind directly to PAK3, blocking its production, and thus SMAD4 deletion activates the PAK3-JNK-Jun pathway, which in turn accelerates cancer metastasis [107]. Regulator of G protein signalling 6 (RGS6) inhibits the formation of complexes between SMAD4 and SMAD2/3 by binding to SMAD4, inhibiting the TGF-β pathway and thus inhibiting EMT in NSCLC [108]. In addition, reduced SMAD4 expression is more common in NSCLC in patients with poor disease-free survival and resistance to platinum-based chemotherapy. SMAD4 mutations are an independent risk factor for survival in NSCLC patients. SMAD4 mutations or deletions and reduced expression can be used to identify NSCLC patients with poor survival and resistance to platinum-based chemotherapy and are predictive markers or therapeutic targets for NSCLC [109].

SMAD7 affects the progression of lung cancer mainly by influencing EMT, a complex nonlinear biological process. A hallmark of EMT is the switch-like behaviour during state transitions, which are characteristic of phase transitions. Therefore, the detection of critical points prior to interstitial state transitions is essential for understanding the molecular mechanisms of EMT. Through dynamic network biomarker (DNB) modelling, SMAD7 was identified as one of the DNB genes that can promote EMT by switching the regulatory network of SMAD7. Survival analysis revealed that SMAD7 further acts as a prognostic biomarker for lung adenocarcinoma as a DNB gene [110]. REGγ (a proteasome activator) can activate SMAD7 through degradation of TGF-β signalling and promote lung cancer metastasis [111]. In lung cancer, the protein arginine methyltransferase 5 (PRMT5) binds to SMAD7 and methylates it on arginine-57, enhancing the binding of SMAD7 to the IL-6 co-receptor gp130, thereby ensuring potent activation of STAT3, which in turn promotes tumour cell proliferation and progression [112]. The above studies suggest that both SMAD4 and SMAD7, like SMAD3, can exert pro- and anti-cancer effects through a large mediation network.

### 1.7. Breast Cancer

Breast cancer is the most common malignancy and the leading cause of cancer-related death in women. Although there are many clinical methods to diagnose and treat breast cancer, breast tissue biopsy is still the best method to confirm the diagnosis of breast cancer, and surgical excision is still the main treatment for breast cancer. The exact molecular mechanisms underlying the progression of breast cancer are still unclear, but recent studies have shown that the TGF-β pathway and SMAD proteins play an important role in the diagnosis, treatment, and prognosis of breast cancer.

In breast cancer, the TGF-β pathway mainly exhibits pro-cancer effects. Activation of SMAD3 in the TGF-β pathway is predominant in breast cancer, and a large number of post-translational modifications of SMAD3 play a role in the development of breast cancer. It was found that acetylation of SMAD3 by KAT6A at K20 and K117 promoted the binding of SMAD3 to the oncogenic chromatin modifier triple motif-containing 24 (TRIM24) and disrupted the interaction between SMAD3 and the tumour suppressor TRIM33. This event in turn promotes KAT6A acetylation of the H3K23-mediated recruitment of the TRIM24–SMAD3 complex to chromatin, thereby increasing SMAD3 activation and the expression of immune response-associated cytokines, leading to breast cancer stem-like cell stemness, myeloid-derived suppressor cell recruitment, and enhanced metastasis in triple-negative breast cancer. This study suggests a KAT6A acetylation-dependent regulatory mechanism that controls SMAD3 oncogenic function and provides insight into how targeting epigenetic factors by immunotherapy may enhance antimetastatic efficacy [113].

Furthermore, EZH2 mediates the upregulation of SMAD3-K53/K333 methylation, facilitating the interaction of SMAD3 with its cell membrane localisation molecules, which in turn maintains the phosphorylation of SMAD3 by the TGF-β receptor. This finding reveals a complex layer involved in regulating SMAD3 activation that is coordinated by EZH2-mediated SMAD3-K53/K333 methylation, which drives cancer metastasis [114]. In addition, OSR1 acts directly on SMAD2/3 and phosphorylates 223Thr in the SMAD2-linked region and 179Thr in the SMAD3-linked region, promoting TGF-β pathway signalling and breast cancer metastasis [115]. In triple negative breast cancers, growth differentiation factor-10 (GDF10), a member of the TGF-β superfamily, was downregulated in tumour samples. Overexpression of GDF10 inhibited proliferation, invasion, and epithelial mesenchymal transformation by upregulating SMAD7 and E-Cadherin, downregulating p-SMAD2 and N-Cadherin, and reducing nuclear SMAD4 expression [116].

At the same time, SMAD proteins were found to act as targets for a variety of breast cancer drugs. In triple negative breast cancer, Asiaticoside inhibited the expression of TGF-β1 and phosphorylation of SMAD2/3, suppressing the development of triple negative breast cancer [117]. Trx-SARA is an example of a peptide aptamer that links to SMAD2 and SMAD3, disrupting their communication with SMAD4. Trx-SARA treatment of mouse mammary epithelial cells reduced the SMAD2 and SMAD3 levels and inhibited TGF-β-stimulated EMT [118]. Salvianolic acid B inhibits the TGF-β pathway through downregulation of SMAD2/3/4 expression, which in turn inhibits the activation of tumour-associated fibroblasts and reshapes the tumour microenvironment [119]. In addition, the pro-cancer effects of SMADs provide targets for the development of future breast cancer drugs. In breast cancer, breast tumour kinase (BRK) phosphorylates SMAD4 and promotes its recognition by the ubiquitin-proteasome, thereby accelerating the degradation of SMD4. Activated BRK-mediated SMAD4 degradation is associated with suppression of the tumour suppressor gene FRK and increased expression of mesenchymal markers, SNAIL and SLUG. Combination therapies targeting activated BRK signalling may therefore have a synergistic effect in the treatment of SMAD4-deficient cancers [120].

SMAD7, a suppressor molecule of the TGF-β pathway, is strongly associated with the prognosis of breast cancer patients. In breast cancer, SMAD7 expression is decreased by the overexpression of SET domain divergent histone lysine methyltransferase 1 (SETDB1), while knockdown of SETDB1 upregulates SMAD7 expression, thereby inhibiting breast cancer metastasis, suggesting that increased SMAD7 expression improves patient survival [121]. Furthermore, in breast cancer, the BMP4–SMAD7 signalling axis blocks tumour metastasis, and increased levels of BMP4 and SMAD7 predicted improved recurrence-free survival and OS in breast cancer patients [122]. In summary, SMAD proteins play an important role in the progression of breast cancer by assisting in the activation of the TGF-β pathway. With in-depth studies on the TGF-β pathway and SMAD proteins, SMAD proteins are expected to become important biomarkers for breast cancer diagnosis and provide therapeutic targets for breast cancer, especially for triple negative breast cancer with poor prognosis.

### 1.8. Ovarian and Uterine Tumours

Among the female germline tumours, ovarian and uterine tumours are significantly influenced by the TGF-β pathway and SMAD proteins. In ovarian cancer, SMAD1 positively regulates SOX2 and SMAD3 negatively regulates SOX2, which acts as a regulatory node that controls the signalling of the TGF-β pathway, which in turn affects ovarian cancer metastasis [123]. In addition, SMAD3 upregulation, by inducing STYK1 expression, promotes cell invasion migration and the EMT process and enhances tumour cell resistance to paclitaxel [124]. In ovarian granulosa cell tumours, the mutant protein FOXL2C^134W^ was highly expressed. A novel hybrid DNA motif, AGHCAHAA, unique to the FOXL2C^134W^ mutant, gained the ability to bind SMAD4, forming the FOXL2C^134W^/SMAD4/SMAD2/3 complex. This binding induced an enhancer-like chromatin state, leading to the transcription of nearby genes, many of which are characteristic of epithelial-to-mesenchymal transition. It promotes the ability of ovarian granulosa cell tumours to migrate aggressively. FOXL2C^134W^ and its interaction with smad4 are potential therapeutic targets for this disease [125].

In uterine tumours, TGF-β, on the other hand, exhibited predominantly tumour suppressive effects. Kriseman et al. identified a critical role for SMAD2/3 in maintaining normal endometrial function and confirmed the hormone-dependent nature of SMAD2-3 in the uterus. Dual conditional inactivation of SMAD2 and SMAD3 in the mouse uterus leads to endometrial dysregulation, sterility, and ultimately bulky endometrioid uterine carcinoma [126]. Furthermore, Huang et al. found that SMAD2/3 signalling controls endometrial regeneration and differentiation and that mice with SMAD2/3 conditional deletion develop metastatic uterine tumours and that SMAD2/3 inactivation affects the morphology and differentiation of uterine-like organs. Among the 849 patients with endometrial cancer, 52 patients had mutations in SMAD2 including 41 missense mutations and 11 truncating mutations. Forty-one patients had mutations in SMAD3 and 36 patients had mutations in SMAD4 [127]. The tumour suppressor function of SMAD2/3 in the endometrium was found to be mediated by the PI3K/PTEN/AKT signalling pathway, which has an important role in regulating epithelial cell homeostasis and interfering with the effect of both pathways on endometrial carcinogenesis. Studies have shown that PTEN inhibits the nuclear translocation of SMAD2/3 and that nuclear translocation of SMAD2/3 inhibits PTEN deficiency-induced tumourigenesis [128].

In addition, a clinical study found that single nucleotide polymorphisms (SNPs) in SMAD2 (rs4940086 and rs8085335) have an impact on the development of cervical cancer risk in the Bangladeshi population. The association of cervical cancer susceptibility with selected SNPs was assessed by multiple logistic regression. Rs4940086 heterozygous genotype (T/C) of SMAD2 was associated with a 3.89-fold higher risk of developing cervical cancer (*p* = 0.001, AOR 3.89, 95% CI 1.777–8.513). The combination of T/C and C/C genotypes also significantly elevated the risk of cervical cancer (*p* = 0.035, AOR 1.876, 95% CI 1.047–3.364). The results suggest that polymorphic variants of SMAD2, particularly rs4940086, may increase cervical cancer susceptibility in Bangladeshi women [129]. According to the above findings, unlike other malignancies, the TGF-β pathway plays a predominantly tumour suppressive role in uterine tumours, especially SMAD2/3. Therefore, by comparing the differences in TGF-β pathway-related molecules in uterine tumours and other cancers, it is expected to find the key regulatory mechanisms of the TGF-β pathway’s role in tumours.

### 1.9. Other Cancers

In haematological neoplasms, aberrant activation of the BMP2/SMAD pathway during the transformation of myeloproliferative neoplasms to leukaemia leads to aberrant self-renewal of megakaryocytic red lineage podocytes [130]. In chronic lymphocytic leukaemia(CLL), the expression of SMAD1, SMAD5, and SMAD8 is significantly increased and their overexpression is associated with short patient survival [131]. In acute myeloid leukaemia (AML), 84 patients with de novo AML receiving chemotherapy and 71 patients receiving allogeneic haematopoietic stem cell transplantation (allo-HSCT) were analysed and the Kaplan–Meier survival assessment showed that high expression of both SMAD3 and SMAD7 was associated with poor event-free survival (EFS) and OS; high expression of SMAD6 was associated with shorter EFS and OS in allogeneic HSCT patients. Multifactorial analysis showed that only high SMAD7 expression had an independent adverse effect on EFS and OS and was a better prognostic marker than SMAD3 [132]. In chronic myeloid leukaemia (CML), oncogenic BCR-ABL1 and the cellular ABL1 tyrosine kinase phosphorylate and inactivate SMAD4, thereby blocking the ability of SMAD4 to activate cell cycle protein-dependent kinase (CDK) inhibitor expression and induce cell cycle arrest. Inhibition of BCR-ABL1 kinase by imatinib prevents SMAD4 tyrosine phosphorylation and re-sensitises CML cells to TGF-β-induced anti-proliferative and pro-apoptotic responses [133].

In melanoma, SMAD7 expression was assessed by immunohistochemistry in 205 cutaneous melanoma primary tumours and the results were correlated with the clinicopathological features of the patients. High SMAD7 expression was positively correlated with several features of tumour aggressiveness such as the presence of ulceration (*p* < 0.001), higher tumour thickness (*p* < 0.001), and mitotic rate (*p* < 0.001), but not the presence of regional or distant metastases. Furthermore, high SMAD7 expression independently predicted unfavourable outcomes: melanoma-specific survival (hazard ratio = 3.16, *p* < 0.001) and recurrence-free survival (hazard ratio = 2.88, *p* < 0.001). SMAD7 was defined as a marker of aggressive tumour behaviour and poor clinical outcome in melanoma patients [134]. In addition, the 26S proteasomal non-ATPase regulatory subunit 14 (PSMD14) could inhibit melanoma growth and migration through SMAD3 accumulation or SLUG reduction, respectively [135]. MED1 low expression promoted the TGF-β signalling pathway by inhibiting SMAD2 ubiquitination to facilitate the cellular transition from epithelial to mesenchymal phenotype and migration [136].

In addition, SMAD proteins play an important role in urological tumours, particularly prostate cancer. It also has a significant impact on the development of head and neck tumours and osteosarcoma, as detailed in Table 1. In addition to those reviewed here, a variety of non-coding RNAs can also influence the biological behaviour of tumours by affecting SMAD proteins, as detailed in Table 2.

**Table 1 diagnostics-13-02769-t001:** The role of SMAD proteins in other tumours.

Tumours	Upstream Molecules	Target Molecules	Biological Impact	Ref.
Nasopharyngeal Carcinoma	TIM-3	SMAD2 and SMAD7	Promoting EMT	[137]
HPV-positive head and neck tumours	HPV	SMAD4	Promoting DNA damage repair Promoting viral replication and tumourigenesis Enhancing resistance to cisplatin	[138]
SMAD4-deficient squamous cell carcinoma of the head and neck	Olaparib		Promoting DNA damage and reducing proliferation	[139]
Renal cell carcinoma	Calcitriol	Enhanced interaction of vitamin D receptors with SMAD3	Inhibiting the migration and invasion	[140]
Prostate cancer	HNF1B	SMAD6	Inhibiting tumour proliferation	[141]
	SAMD7	c-JUN and HDAC6	Promoting aggressive migration of prostate cancer	[142]
	PLK1 and PELO	SMAD4	Promoting proliferation and metastasis	[143]
	HOXD13	SMAD1	Inhibiting EMT	[144]
	SMAD3	AR	Promoting castration tolerance in prostate cancer	[145]
	Sasanquasaponin	SMAD2/3	Inhibiting the aggression of prostate cancer	[146]

**Table 2 diagnostics-13-02769-t002:** Interaction of non-coding RNAs with SMAD proteins in tumours.

Tumours	Upstream Molecules	Downstream Molecules	Target Molecules	Biological Impact	Ref.
Gastric cancer	MicroRNA-135		SMAD2	Suppressing metastasis	[147]
	TMEM147-AS1	miR-326	SMAD5	Promoting invasive metastasis	[148]
	LncRNA-HCP5	miR-299-3p	SMAD5	Inhibiting proliferation, invasion, migration, and promotes apoptosis	[149]
	LncFGD5-AS1	miR-196a-5p	SMAD6	Suppression of EMT	[150]
Colorectal cancer	CircPTEN1		Binds SMAD4 competitively with SMAD2/3	Suppression of EMT	[151]
	LINC00657	SMAD2	HPSE	Accelerating the CRC invasion	[152]
	LINC00941		SMAD4	Promoting invasive migration	[153]
	Circ-SMAD2			Promoting proliferation and invasion	[154]
	miR-186-5p		SMAD6/7	Inhibiting proliferation and migration	[155]
	MiR-581		SMAD7	Promoting CRC metastasis	[156]
Hepatocellular carcinoma	MiR-1258		SMAD2/3	Inhibiting the metastasis of HCC	[157]
	LINC00261		SMAD3	Suppression of EMT	[158]
	LINC01410	miR-124-3p	SMAD5	Promoting invasive migration	[159]
	CircFGGY	miR-545-3p	SMAD7	Inhibits proliferation and invasion of HCC cells	[160]
	MiR-21-3p	SMAD7	YAP1	Promoting HCC invasive migration	[161]
	miR-148a-3p	Argonaute 2	SMAD2	Inhibits migration and proliferation of HCC cells	[162]
Pancreatic cancer	MiR-487a-3p		SMAD7	Suppressing the malignant progression of PC	[163]
Lung Cancer	CircSCAP		SMAD2	Promoting the metastasis of NSCLC	[164]
	OSER1-AS1	microRNA-433-3p	SMAD2	Promoting proliferation and invasive migration in NSCLC.	[165]
	MiR-361-5p		SMAD2	Promoting lung adenocarcinoma	[166]
CLL	miR-26b-5p	SMAD4	C-Myc	Promoting CLL progress	[167]
Adult T-cell lymphoma	LINC00183	miR-371b-5p	SMAD2	Promoting chemotherapy resistance	[127]
Ovarian cancer	CASC15	miR-23b-3p/miR-24-3p	SMAD3	Promoting the development of ovarian cancer	[168]
Endometrial cancer	MCTP1-AS1	miR-650	SMAD7	Inhibiting proliferation invasion, migration, and EMT in endometrial cancer	[169]
	Hsa_cir_0001860	miR-520	SMAD7	Inhibiting cell migration and invasion	[170]
Bladder Cancer	Circ-0002623	miR-1276	SMAD2	Promoting the malignant phenotype of bladder cancer	[171]
	CircNCOR1	HNRNPL	SMAD7	Inhibiting lymph node metastasis from bladder cancer	[172]
	MiR-5581-3p		SMAD3	Inhibiting bladder cancer cell migration and proliferation	[173]
	PlncRNA-1	miR-136-5p	SMAD3	Promoting bladder cancer progression	[174]
	KCNMB2-AS1	miR-3194-3p	SMAD5	Promoting bladder cancer progression	[175]
Prostate cancer	SMAD3	PCAT7	MiR-324-5p	Promoting bone metastases from prostate cancer	[176]
Breast cancer	MiR-135-5p		SMAD3	Inhibiting EMT and breast cancer metastasis	[177]
	ARHGAP5-AS1		SMAD7	Inhibiting stress fibres in breast cancer cells to inhibit cell migration	[178]
Melanoma	NEAT1	MiR-200b-3p	SMAD2	Promoting EMT	[179]
Osteosarcoma	MiR-135a		SMAD2	Promoting proliferative migration	[180]
	LINC00266-1	miR-548c-3p	SMAD2	Promoting osteosarcoma progression	[181]
	MiR-16-5p		SMAD3	Enhancing cisplatin sensitivity	[182]

## 2. Discussion

From clinical cohort studies to the development of mouse models to the study of cells in culture, the dual role of TGF-β in cancer has long been identified, but its mechanistic basis, operational logic, and clinical relevance remain unclear. What causes altered TGF-β signalling in cancer? Which steps of tumour progression may benefit from a defective TGF-β pathway? When does transforming growth factor beta act as a metastatic signal? Most importantly, how is it used to treat cancer? These questions remain the greatest impediment to the development of effective TGF-β-targeted therapies.

To answer the appeal question, researchers have spared no effort in exploring the issue. Current research has identified that the activation rates of SMAD2 and SAMD3 may be one of the mechanisms by which TGF-β produces its dual effects. SMAD2 and SMAD3 are the main downstream effectors of TGF-β, and despite their 84% protein sequence homology and similar activation patterns, there is growing evidence that SMAD2 and SMAD3 play different and opposing functions including cell invasion, tumour growth, and metastasis [183,184,185]. In addition, tumour cells can evade anti-tumour surveillance by TGF-β through the accumulation of mutations in TGF-β signalling and use the defective TGF-β pathway to promote their own development.

At present, many anti-cancer drug ecological interventions targeting TGF-β have undergone pre-clinical and clinical phases. However, given the systemic effects of the TGF-β pathway and its own important physiological functions, systemic administration may disrupt normal physiological functions, and the results of clinical trials have been unsatisfactory. SMAD proteins, as important transcription factors, play irreplaceable roles in delivering and integrating the TGF-β signalling pathway and other signalling networks through a large number of different protein interactions and post-translational modifications. Therefore, the development of small molecule inhibitors of SMAD proteins may make up for the lack of drugs targeting the TGF-β pathway. In addition, given the wide range of actions of SMAD proteins and the large number of target genes, single-cell approaches can be used to identify the origin of the cells that respond to the action of SMAD proteins, gain insights into how specific cell types respond to these signals or adapt to a lack of responsive signals, and explore the precise downstream target genes for each tumour. This will contribute to precision therapy with small molecule drugs.

On the other hand, the function of SMAD proteins in the non-TGF-β pathway cannot be ignored, and some studies have already discovered the role of SMAD proteins in tumours independently of TGF-β. It is believed that with the efforts of more researchers, the functions of the TGF-β pathway and SMAD proteins will be better interpreted, providing more accurate diagnosis and better treatment for tumours.

## Figures and Tables

**Figure 1 diagnostics-13-02769-f001:**
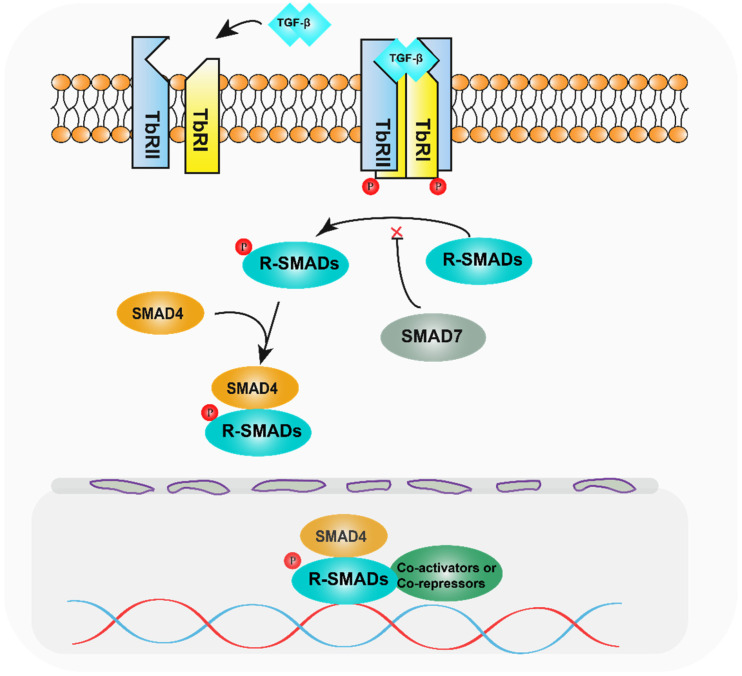
Diagram of the TGF-β pathway mechanism. After TGF-β activates the relevant receptors, the receptors phosphorylate R-SMADs, which bind to SMAD4 and enter the nucleus, recruiting relevant transcription factors and regulating target genes.

## Data Availability

Not applicable.

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
