# Peer review of "SMAD Proteins in TGF-β Signalling Pathway in Cancer: Regulatory Mechanisms and Clinical Applications"

_diagnostics, 2023, doi:10.3390/diagnostics13172769_

Round 1
Reviewer 1 Report
Overall, this is a very comprehensive review of SMADs in various cancers. Although the content is thorough, it can be a little dense to read with so many facts in large paragraphs. I would suggest breaking up large paragraphs (one example is the final paragraph in the Discussion section). Additionally I have the following suggestions:
- “SMADs proteins” should be “SMAD proteins”. This should be changed throughout the manuscript
-Uniformity of the beta symbol should be used. The authors interchange “TGF-b, TGF-beta, TGFβ, and TGF-β”. I would suggest using “TGF-β” . This should be changed throughout the manuscript
-Page 1, last paragraph, “depapentaplegic” should be “decapentaplegic”. This same sentence is missing a period at the end.
-“Caenorhabditis elegans” and “Drosophila melanogaster” should be italicized. Please check throughout the manuscript
-Page 2, first full paragraph, “Eight Smad proteins are encoded”, “Smad” should be “SMAD”. Same with the last word on Page 1. Please check consistency throughout the entire manuscript. Sometimes “Smad” is used, other times “SMAD” is used, both when discussing proteins.
- Page 2, first full paragraph “TGF family receptor” should be “TGF-β family receptor”
-Page 2, first full paragraph “anti-Miller receptors should be “anti-Mullerian hormone receptors”
- Page 2, first full paragraph “SMAD2 and 3 principally for the TGFβ, activin, and Nodal receptors” should be “SMAD2 and 3 are principally substrates for the TGFβ, activin, and Nodal receptors”
-Change all hyphens to non-breaking hyphens to make it easier to read (e.g. “mod-ification”)
- Page 2, second full paragraph “and the phosphorylation site for protein kinase[6].” Which protein kinase?
-Define “GS structural domain” on page 2 during its first usage
-Remove “the” in “Activated R-SMADs bind to Co-SMAD, the SMAD4, to form a polymer that completes”
-The abbreviation “pSmad2/3L-Thr” should be placed immediately after it’s full name in the esophageal cancer secion “linker threonine-phosphorylated Smad2/3 (pSmad2/3L-Thr)”
- “in situ” should be italicized
-define “OS” on page 5 during first usage
-define “TNM stage” on page 5 during first usage
-Page 5, “(Stat)-3” should be “(STAT)-3”. STAT should be capitalized throughout the manuscript
-“et al.” should be italicized
-Table 1 under RCC, Chinese character is placed “PLK1ĺ’ŚPELO”
Please see comments above
Author Response
Author's Reply to the Review Report
Many thanks you for your valuable comments on our manuscript, which helped us a lot. I have revised the manuscript line by line according to the comments, please criticize and correct me.
First, it is true that reading so many facts in a large paragraph can be quite dense, so I have broken up the large paragraphs as your suggestion. I will respond to other comments point by point below.
- I have changed “SMADs proteins” to “SMAD proteins” throughout the manuscript.
- I have used TGF-β throughout the manuscript.
- I have changed “depapentaplegic” to “decapentaplegic”. I have added the period at the end of this sentence.
- I have changed “Caenorhabditis elegans” and “Drosophila melanogaster” to italicized.
- I have changed “Smad” to “SMAD” throughout the manuscript.
- I have changed “TGF family receptor” to “TGF-β family receptor.”
- I have changed “anti-Miller receptors to “anti-Mullerian hormone receptors”.
- I have changed “SMAD2 and 3 principally for the TGFβ, activin, and Nodal receptors” to “SMAD2 and 3 are principally substrates for the TGFβ, activin, and Nodal receptors”
- I have changed all hyphens to non-breaking hyphens.
- I have supplemented with specific protein kinases, namely TGF‑β type I receptor.
- I have defined the “GS structural domain”.
- I have removed “the” in “Activated R-SMADs bind to Co-SMAD, the SMAD4, to form a polymer that completes”
- I have placed the abbreviation “pSmad2/3L-Thr” after it’s full name in the esophageal cancer secion “linker threonine-phosphorylated Smad2/3 (pSmad2/3L-Thr)”.
- I have changed “in situ” to italicized.
- I have defined “OS” on page 5 during first usage.
- I have defined “TNM stage” on page 5 during first usage.
- I have made sure that all STAT in the manuscript have been capitalized.
- I have changed “et al.” to italicized.
- I have changed the Chinese to English in the table1.

Reviewer 2 Report
The current paper is a review article on the Smad-dependent signaling pathways in cancer progression.
I believe that this review article is helpful to update relevant knowledge.
Overall, the current manuscript is well-organized.
However, I would like to raise the comments as below.
1) The role of Smad1/5 (Smad-dependent BMP signaling pathway) is not mentioned adequately in the present manuscript.
The title of the paper should be modified.
2) In the introduction, activin receptor-like kinase 5 (ALK5) should be written together as another name for TbRI.
3) As far as I know, “On ligand binding, TbRI forms a heterotetramer with TbRII, and TbRI phosphorylates the GS structural domain of TbRII to activate it.” is incorrect.
TbRI and TbRII should be replaced each other.
4) It seems that several important papers are not cited in the current manuscript.
Especially, the authors do not introduce several papers on transcriptional co-activators (p300, CBP, and GCN5) or transcriptional co-repressor (c-Ski and SnoN).
These are important for the regulation of Smad-dependent gene transcription.
5) To my knowledge, Smad6 can inhibit TGF-b signaling only when overexpressed.
Thus, at least in physiological conditions, the effect of Smad6 on tumor progression is thought to be derived from the inhibition of TGF-b signaling.
If the authors focus on the role of Smad-dependent TGF-b signaling, this should be considered (Figure 1 or other relevant descriptions).
6) In the chapter of pancreatic cancer, DPC4 should be written as another name for Smad4.
7) In Table 1, “upstream molecules” and “target molecules” are confusing.
Smads are listed in the upstream in some cancers, whereas Smad are also targets in others.
Author Response
Author's Reply to the Review Report
Thank you very much for your valuable comments on our manuscript, which are very helpful. I have revised the manuscript point-to-point according to the comments, and I would be grateful for your criticism.
- As you say, our manuscript lacks elaboration on SMAD1/5, so I've changed the title to “SMAD proteins in TGF-β signalling pathway in cancer: regulatory mechanisms and clinical applications”.
- I have written ALK5 together as another name for TbRI.
- It's our fault. I have replaced TbRI and TbRII with each other.
- We have added content and references about SMAD co-activators and co-repressors in the fifth paragraph of the introduction.
- We have added content about the differences between SMAD6 and SMAD7 in the TGF-β pathway in the third paragraph of the introduction. “Smad6 preferentially inhibits BMP signalling, whereas Smad7 inhibits both TGF‑β/activin and BMP signalling pathways.” And we have removed SMAD6 in Figure1.
- We have written DPC4 together with SMAD4 in the chapter of pancreatic cancer.
- Because according to the references, in some cancers, SMADs act as upstream molecules by regulating other molecules in the cancer. And in some others, other molecules function by regulating SMADs. Thus, SMADs are listed in upstream molecules in some tumors and in target molecules in others.

Reviewer 3 Report
Comments to the authors:
The review entitled “SMADs proteins in cancer: regulatory mechanisms and clinical applications” is interesting and important in terms of the role of SMADs in cancer progression and their potential for being prognostic or diagnostic biomarkers.
Although the authors have made a good attempt to compile the information, the manuscript is not well written. It lacks flow and continuity in elaborating on the topics and is unnecessarily lengthy in a few places. There are multiple grammatical errors and ambiguous and out-of-context sentences. Proper referencing is also required.
Major overhauling of the manuscript is required. I want to point few examples out of many for the authors to get an idea and correct the whole manuscript accordingly:
1. In the abstract, reframe the sentence “The complex roles and important functions of SMADs proteins lead to their roles and mechanisms in tumors remain a hot topic in current oncology research.”, as it is not clear or misleading.
2. In the introduction section, break the first sentence into smaller sentences. Also, the first few sentences lack references. Proper references should be provided.
3. Put “.” after this sentence “The first protein of the SMAD family, an intracellular protein named Mad, was identified in the 1990s during a mutation screen for the depapentaplegic (dpp) gene responsible for the formation of Drosophila wings[4]”.
4. Reframe “This discovery paved the way for the discovery of a whole new family of proteins, the SMAD family” the sentence.
5. Change all the scientific names into italics.
6. Elaborate on BMP before abbreviating it, in the sentence “SMAD1, 5, and 8 are mainly substrates for BMP and anti-miller receptors;”. There are other places where abbreviations are used without first mentioning the full name. Kindly correct them.
7. Eight Smad proteins are encoded in human and mouse genomes, correct “encoded in” to “encoded by”.
8. There is the repetition of 2 sentences “Smad family proteins function primarily in the TGFβ pathway. SMADs family proteins function primarily in the TGFβ pathway.” Also, there should be uniformity in writing the protein name while using upper or lower case.
9. Abbreviate the words at their first use before using the abbreviation. Example BE for Barrett's esophagus.
10. Provide a reference for the first paragraphs of Esophageal and gastric cancer.
11. The last part of the gastric cancer section seems to be copied “Based on previous studies, it is known that the TGF-β pathway promotes the EMT process in tumours and is often associated with a poorer prognosis. However, the above data suggest that impaired TGF-β signaling is also associated with poor prognosis in some patients, and the complexity of whether TGF-β signaling is tumour suppressive or tumour enabling remains to be further investigated.”. This should be corrected to clear the ambiguity of these sentences.
12. “For SMAD7, which has a dual role in CRC progression, SMAD7 promotes tumourigenicity in non-tumourigenic CRC cell lines by inhibiting the TGF-β signalling pathway[35].” Repetitive use of SMAD7 should be avoided.
13. Unnecessary and out-of-context sentences like “In addition to SMAD4, recent studies have found that FAM198B can pro-mote CRC progression by targeting SMAD2 to regulate the M2 polarisation of macrophages[40].”, Should be removed. What is FAM198B? Authors should not write anything without a little background.
14. In the CRC section, SMAD4 is first discussed in the beginning, then SMAD7 and SMAD3 and then again SMAD4. Avoid discussing the same protein at multiple places in the same section and also avoid unnecessary elaboration of the topic to make the review lengthy.
15. The discussion section is more focused on the TGF-β, while the authors are supposed to discuss more about the role of SMADs. The authors should discuss the SMADs and their possible drug targeting, clinical significance, and prospects in cancer therapy.
English language is not up to the standard and requires extensive editing.
Author Response
Author's Reply to the Review Report
Thank you very much for your valuable comments on our manuscript, which are very helpful. I have revised the manuscript point-to-point according to the comments, and I would be grateful for your criticism.
I have re-organized the structure of the manuscript and deleted some parts that were not sufficiently related to the topic of the article. I have also corrected the grammatical errors and added the proper references. Thank you very much for your comments.
- I have changed the sentence to “The roles and mechanisms of SMADs proteins in tumours remain hot topics in current oncology research because of their complex roles and important functions. ”
- I have broken the first sentence into smaller sentences, and provided the proper references.
- I have put “.” after this sentence “The first protein of the SMAD family, an intracellular protein named Mad, was identified in the 1990s during a mutation screen for the depapentaplegic (dpp) gene responsible for the formation of Drosophila wings”.
- I have deleted the sentence because it is not necessary for the context.
- I have changed all the scientific names into italics.
- I have elaborated on BMP before abbreviating it.
- I have corrected “encoded in” to “encoded by”.
- I have deleted the repetition of the sentences. I have standardized the case of protein names.
- I've added the full name before the first use of the abbreviation.
- I have provided references for the first paragraphs of Esophageal and gastric cancer.
- I have corrected these sentences.
- I have reframed the sentence.
- I have removed sentence.
- I have re-ordered the paragraph and removed redundancies.
- I have removed the excessive TGF-beta content and added more discussion of SMAD.

Round 2
Reviewer 3 Report
In the revised version, the authors have made reasonable efforts to improve the quality of the manuscript. A couple of minor corrections are needed.
In the abstract, correct the sentence as: "The complex role, mechanisms, and important functions of SMAD proteins in tumours are the hot topics in current oncology research.”
In the sentence “Smad6 preferentially inhibits BMP signalling, whereas Smad7 inhibits both TGF-ββ/activin and BMP signalling pathways”, write the SMAD in upper-case letters.
There are still some minor errors in the manuscript which authors are requested to take care of before finally submitting the manuscript.
Minor errors were found.
